# Token-efficiency based Routing Technique within Mixture of Experts Architecture for Large Language Model

## Abstract

Mixture-of-Experts (MoE) architectures have emerged as a powerful technique for improving and scaling Large Language Models by conditionally activating Feed Forward subnetworks and distributing tokens through a routing system, within the Transformer layers. However, existing MoE methods often rely on static top-k routing strategies that do not involve token-level variability in complexity, leading to suboptimal expert utilization. In this research, we propose a novel token-complexity-based routing framework that dynamically allocates tokens to either lightweight or strong feedforward networks (FFNs) based on their estimated token complexity. Our router is trained using a few-shot classification objective to distinguish between easy and complex tokens and a surrogate neural network layer. The efficacy of the framework is evaluated while integrating the router with Mistral-7B, Mixtral-8×7B and Llama-2-7B model. We evaluate our approach on several benchmarks from various fields, and our proposed MoE framework improves accuracy up to 12% compared to the state-of-the-art results using different MoE architecture, with reasonable computational cost.

## 1 Introduction

Large Language Models (LLMs) have achieved decent performance across a broad range of Natural Language Processing (NLP) tasks encompassing text generation, question answering, and solving intricate math and coding problems. However, recent architectures such as GPT-4, Claude, PaLM, and Mistral have scaled to billions of parameters, which leads to high energy consumption for their increased computational costs, inference latency, and resource requirements Jegham et al. (2025). It also suffers from hallucinations while answering intricate questions that involve problem-solving. Mixture-of-Experts architecture has emerged to make a solution, allowing dynamic activation of subsets of model components, typically Feedforward Networks (FFNs) within the transformer layer. Activation of a subset of FFN enables parameter growth without the proportionate increase in computational cost, and keeps the accuracy of the model stable. In this research, we are evaluating the effect of MoE Architecture on the accuracy of the models' performance while keeping the computational cost stable of the fine-tuned models' compared to the base models. We use a Mixture-of-Experts (MoE) architecture with an efficient Gating Function/Router that distributes the input tokens to the Feed Forward Networks of the Transformer layers based on the capacity of the FFNs and the complexity of the tokens. Our proposed router is lightweight and fully pluggable to any state of the art MoE-based LLM. In this paper, we will be referring to the gating mechanism as the transformer's Router and vice versa.

In MoE architecture, a top-k gating mechanism is employed Zhou et al. (2022), where only a fixed number of experts (also known as FFN submodules) are activated per token. This method is effective while reducing computational cost, but somewhat relies on simplistic heuristics or softmax-based gating mechanisms, lacking the capacity to reason about the actual difficulty or complexity of input tokens, yielding a compromise in performance. As a result, these models may misallocate tokens to experts, such as easy tokens are being handled by intricately designed FFN and vice versa, limiting their efficiency and precision. On the other hand, creating experts with similar intricate architecture risks the pitfall of higher computational cost.

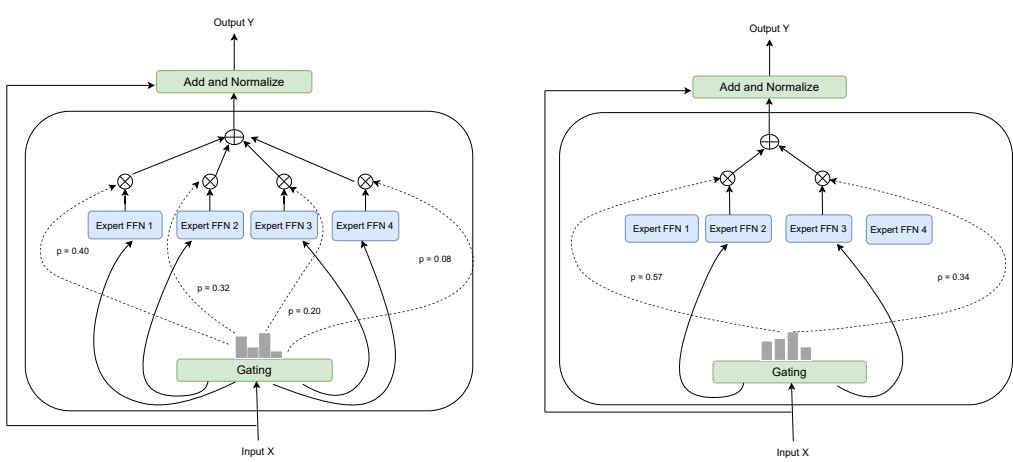

Figure 1: Transformer layer with proposed router and optimal load load balancing

(a) Dense MoE                                    (b) Sparse MoE

Figure 2: Comparison of dense versus sparse Mixture-of-Experts (MoE) layers.

In this work, we introduce a complexity-aware routing framework that enhances the feedforward sublayer of transformer models. Our method leverages token-level difficulty estimation to determine whether a token should be routed to a lightweight or a strong FFN. The router is trained using a few-shot setting and a neural network layer. Tokens deemed "easy" are processed by two lightweight FFNs, while "hard" tokens receive more expressive computation from two other stronger FFNs, allowing the model to dynamically balance computational cost and accuracy.

The novel contributions of our work are summarized as follows,

- We implement a token-complexity-based router for FFN layers of the transformer that dynamically allocates tokens to lightweight and strong experts (FFNs) based on their complexity using a threshold.

- A surrogate neural network layer with few-shot training is added to each fine-tunable transformer layer for appropriate token distribution while using perplexity scoring to determine the models' stability for different activation configuration.

- The proposed lightweight fully pluggable router is integrated with Llama, Mistral, Mixtral transformers layers and performance is validated on several diverse benchmarks.

Table 1: State-of-the-art Mixture-of-Experts (MoE) methods comparison with proposed token-complexity routing

| Method | Contribution | Routing | LLMs Used | Limitations |
|---|---|---|---|---|
| Switch Transformer Fedus et al. (2022) | Sparse MoE with top-1 expert selection | Softmax gating with aux losses | T5-Base, T5-Large | Static routing and less efficient for fine tuning |
| Expert Choice Zhou et al. (2022) | Experts select tokens for load balance based on their semantic complexity | Reverse gating with balance loss | T5-Large | Tokens are chosen by experts which creates sparsity among the experts and process similar tokens |
| ST-MoE Zoph et al. (2022) | Stable sparse MoE training for transfer | Refined losses/capacity control | Vision Transformer (ViT), T5 variants | Focus on stability, not semantic difficulty; no efficiency-specialized experts |
| Mixtral 8×7B Jiang et al. (2024) | Production-grade MoE LLM with 8 experts | Top-$k$ routing with load balance | Mixtral-8x7B | Heuristic routing; no explicit complexity classifier; closed-weight deployment |
| **Proposed** | Light and strong FFNs; few-shot-trained router | Dynamic token complexity within MLP layer | Mistral-7B, Mixtral-$8 \times 7B$, LLaMA-2-7B | - |

## 2 RELATED WORKS

For preserving the performance of LLMs while maintaining the computational efficiency, different MoE architectures are being used, such as post-training adaptation, low rank adaptation, routing mechanism, etc., that selectively activate models' components to maintain the balance between cost and accuracy Cai et al. (2024b). Table. 1, shows the state of the art MoE architecture with their routing process with base LLM and their limitations compared to the proposed framework.

Cai. et. al. proposed Flextron Cai et al. (2024a), which generalizes the idea of elastic networks by embedding MoE-like components into both the Multi-layer Perceptron (MLP) and attention layers using a nested structure. Flextron's router training depends on a surrogate model to estimate downstream loss and guide router updates, which introduces indirect supervision and potential instability. Our work addresses the optimal distribution of tokens within appropriate experts by targeting token-level semantic difficulty directly within the feedforward pathway and simplifying router training using lightweight classifiers trained with few-shot learning and complexity thresholding.

RouteLLM Ong et al. (2024) addresses routing across LLMs such as GPT-4 and Mixtral using human preference data. Their framework effectively balances cost and quality but is designed for inter-model routing at the query level using a threshold, not for intra-model routing at the token level within a single LLM whereas our model offers a lightweight solution that routes tokens to internal experts (FFNs) while operating inside a singular LLM. This improves efficiency without introducing significant routing delays during inference. Similarly, RouterBench Hu et al. (2024) proposes a benchmark for multi-LLM routing systems, formalizing the evaluation of cost-performance trade-offs across various model combinations. The framework focuses on choosing a subset of the LLM ensemble for achieving higher accuracy while keeping the cost minimal. Although these frameworks are created based on accessing the models from a surface level, our idea of using optimal subnetworks for optimal accuracy while maintaining the complexity is inspired by their architecture, such as focusing on query-level decision-making using a threshold within the MLP layer and fine-tuning process. Our work fills this gap by embedding routing directly into the transformer feedforward layers, enabling architectural-level efficiency gains and introducing complexity-aware computation for token distribution within the model.

Finally, Kumari et. al. Nishu et al. (2025) propose a post-training method, DynaMoE, to transform a dense LLM into a Mixture-of-Experts (MoE) model by statically partitioning the feedforward layers into nested subnetworks. This framework resembles the closest approach to ours for distributing tokens based on their difficulty level using the router within the transformer layer. However, the router is trained using derived similarity-based labels, which may misrepresent token complexity as it does not include any true labels for training, and the routing is fixed at deployment time based on threshold sensitivity without contextual flexibility across tasks or layers. In contrast, our method introduces few-shot supervised routers that learn to route based on more semantically informed complexity signals, and supports per-token dynamic routing during inference.

# 3 METHODOLOGY

Our proposed MoE architecture introduces a complexity-aware, token-level routing mechanism for FFNs, trained with a few-shot supervision strategy that enables dynamic, efficient, and scalable inference within a single transformer model and can be integrated with any LLM regardless of their architecture due to its lightweight. The detailed methods of our research are briefly described in the following sections. Figure 1 shows the overall workflow diagram of the Transformer using a complexity-based router. Our approach uses the architecture of the base attention layers and performance of Mistral and Llama are evaluated while maintaining a specific number of active layers while using the proposed router which ensures the models' stability to move forward with implementation and evaluation.

Table 2: Perplexity comparison of Mistral and Llama-2 under different active parameter configurations during training and inference.

| Models | Active parameters | | Perplexity |
| --- | --- | --- | --- |
| | Training | Inference | |
| **Mistral** | 8-12 | 8-12 | 33.5 |
| | 6-12 | 6-12 | 16.6 |
| **Llama-2** | 8-12 | 8-12 | 18.78 |
| | 6-12 | 6-12 | 7.40 |

Table 3: Ablation study of the proposed framework on different benchmarks

| Base Models | Cost (params.) | Router | GSM8K | MMLU | MBPP | Natural Questions | Token Complexity (C/S) |
| --- | --- | --- | --- | --- | --- | --- | --- |
| Mistral-2-7B | ~210M | base | 58.4 | 70.6 | 47.5 | 28.8 | – |
| | | Proposed $\theta = 0.50$ | 56.7 | 72.5 | 47.0 | 31.5 | 71 405 / 2 503 |
| | | Proposed $\theta = 0.482$ | 56.7 | 74.8 | 41.0 | 24.6 | 70 798 / 3 110 |
| | | Proposed $\theta = 0.478$ | 42.8 | 65.2 | 40.2 | 30.0 | 69 343 / 4 565 |
| Llama-2-7B | ~33.6M | base | 82.6 | 45.3 | 20.8 | 26.0 | – |
| | | Proposed $\theta = 0.50$ | 95.0 | 56.4 | 15.0 | 30.2 | 70 410 / 2 498 |
| | | Proposed $\theta = 0.482$ | 88.6 | 50.3 | 20.6 | 30.0 | 70 408 / 3 500 |
| | | Proposed $\theta = 0.478$ | 80.2 | 50.0 | 14.0 | 30.0 | 69 320 / 4 588 |

Table 4: Evaluation time (in GPU hours) across different benchmark inferences for Mistral and Llama-2 using base and the proposed router-based approach.

| Base Models | Router | GSM8K | MMLU | MBPP | Natural Questions | Avg. GPU hrs. |
| --- | --- | --- | --- | --- | --- | --- |
| Mistral | Base | 2.0 | 3.2 | 3.5 | 4.4 | 3.28 |
| | Ours | 2.38 | 3.28 | 4.34 | 4.52 | 3.63 |
| Llama-2 | Base | 3.4 | 2.44 | 5.2 | 4.2 | 3.81 |
| | Ours | 3.8 | 2.5 | 5.2 | 4.8 | 4.07 |

## 3.1 BENCHMARK DATASET

To evaluate the effectiveness and performance of the proposed complexity-based routing framework, we employ several diverse and widely used benchmark datasets, such as GSM8K-5 Shot (Grade School Math 8K) Cobbe et al. (2021), MMLU-2 Shot (Massive Multi-task Language Understanding) Hendrycks et al. (2020), NQ (Natural Questions) Kwiatkowski et al. (2019), and MBPP (Mostly Basic Python Programming) Austin et al. (2021), ARC-c/ARC-e (AI2 Reasoning Challenge, Easy and Challenge Set) Huang et al. (2022), LAMBADA (LAnguage Modeling Broadened to Account for Discourse Aspects) Paperno et al. (2016), PIQA (Physical Interaction QA) Bisk et al. (2020), WinoGrande (Winograd Schema Challenge – Grande Scale) Sakaguchi et al. (2021), SciQ (Science Questions) Auer et al. (2023), and HellaSwag (Highly Entailed Long Language Answers for Situations With Adversarial Generations)Zellers et al. (2019) .

## 3.2 MIXTURE OF EXPERTS (MOE)

MoE neural network architecture is a sparsely activated neural network design that enables conditional computation by selecting a subset of specialized sub-networks, or experts, for each input

Table 5: Iteration-level evaluation time for Mistral and Llama-2 models under base and our approach.

| Base Models | Router | GSM8K | MMLU | MBPP | Natural Languages | Avg. Iteration Time |
|---|---|---|---|---|---|---|
| Mistral | Base | 1.38 | 3.95 | 5.08 | 5.6 | 4.00 |
| | Proposed | 2.04 | 4.06 | 7.29 | 6.84 | 5.05 |
| Llama-2 | Base | 8.2 | 1.16 | 9.27 | 6.6 | 6.3 |
| | Proposed | 10.28 | 1.30 | 9.14 | 7.38 | 7.03 |

Table 6: Accuracy (%) Comparison of Proposed Router with Base and SOTA technique

| Model | Router | Cost | ARC-e | LAMBADA | PIQA | WinoGrande | SciQ | HellaSwag | ARC-c | Token Comp. (C/S) |
|---|---|---|---|---|---|---|---|---|---|---|
| Mistral | Base | – | 80.2 | 75.1 | 80.8 | 75.5 | 96.4 | 61.4 | 50.5 | – |
| | DynaMoE | 10B | 75.0 | 71.0 | 78.3 | 71.8 | 74.0 | 95.2 | 41.5 | – |
| | Proposed $\theta = 0.5$ | ∼210M | 71.0 | 68.0 | 82.0 | 75.0 | 72.0 | 91.0 | 45.0 | 36 444 / 10 361 |
| | Proposed $\theta = 0.482$ | ∼210M | 75.0 | 71.0 | 80.0 | 75.0 | 86.0 | 95.0 | 45.0 | 35 904 / 10 901 |
| | Proposed $\theta = 0.478$ | ∼210M | 68.0 | 66.0 | 70.0 | 65.0 | 78.0 | 90.0 | 42.0 | 1 891 / 44 914 |
| Llama-2 | Base | – | 75.1 | 71.5 | 77.5 | 69.1 | – | – | – | – |
| | Proposed $\theta = 0.5$ | ∼33.6M | 72.0 | 71.5 | 76.0 | 65.0 | 90.0 | 56.5 | 42.0 | 36 563 / 10 242 |
| | Proposed $\theta = 0.482$ | ∼33.6M | 72.0 | 73.0 | 75.0 | 67.0 | 91.0 | 57.0 | 46.0 | 35 283 / 11 522 |
| | Proposed $\theta = 0.478$ | ∼33.6M | 68.0 | 70.0 | 66.0 | 62.0 | 88.0 | 50.0 | 38.0 | 1 891 / 44 914 |
| Mixtral-8 × 7B | Base | – | 90.0 | 67.0 | 84.0 | 71.0 | 90.0 | 64.0 | 85.0 | - |
| | Proposed $\theta = 0.5$ | ∼210M | 92.0 | 56.0 | 85.0 | 73.0 | 90.0 | 65.0 | 86.0 | 44,879/555 |
| | Proposed $\theta = 0.482$ | ∼210M | 91.0 | 56.0 | 85.0 | 74.0 | 92.0 | 65.0 | 86.0 | 33,869/17,329 |
| | Proposed $\theta = 0.478$ | ∼210M | 92.0 | 54.0 | 85.0 | 74.0 | 93.0 | 65.0 | 86.0 | 41261/4,173 |

token. It combines the outputs of multiple expert networks using a learned routing mechanism. Generally, there are two types of MoE architecture, dense Pan et al. (2024) and sparse MoE Riquelme et al. (2021). Figure. 2(a) shows the architecture of Dense MoE. In sparse MoE, instead of activating all model parameters for every forward pass as in dense transformers, MoE models dynamically choose a small number of experts (typically top-k) from a larger pool, thereby increasing model capacity without a proportional rise in computation. This concept is discussed in detail in the next section.

Input text of transformer layer is converted into tokens, which can be base words, subwords, or characters. Different tokens carry varying amounts of semantic information. Common words might be single tokens, while rare words or technical terms might be split into multiple base sub-word tokens, affecting how efficiently the model can represent and process different concepts. To make a solution of this problem, we are using a token complexity calculator for determining token's level of complexity and then passing through an appropriate expert depending on its score using the following equation. We use sigmoid activation function while calculating the accuracy with the prediction entropy of hidden state of token t from the prediction (MLP) sub-layer and attention sub-layer with contextual difficulty features $D_{\text{ctx}}(t)$ and token type embedding $T_{\text{type}}(t)$.

$$C(t) = \sigma\big(W[\,H_{\text{pred}}(t),\, H_{\text{attn}}(t),\, D_{\text{ctx}}(t),\, T_{\text{type}}(t)\,]^{\top} + b\big) \tag{1}$$

Here, pronouns, articles have lower complexity score and considered as easy tokens which does not require expensive processing whereas complex tokens such as nouns, verbs require broader contextual understanding. However, in this reasearch, we are labeling the complex tokens using few-shot learning. Any token consisting complex mathematical symbols (such as logarithms, functions related tokens rather than basic arithmetic symbols), programming syntax are considered as complex token inputs. Traditionally tokens which are not predictable using prior context are considered as complex tokens but here we are not considering those as complex tokens rather focusing on logical complexity of input. Thus we are defining the complexity of each token and pass to the appropriate expert.

Given an input $x \in \mathbb{R}^d$, a set of $N$ expert functions $\{E_1, E_2, \ldots, E_N\}$, and a gating function/router $G(x)$ that produces weights $\{w_1(x), w_2(x), \ldots, w_N(x)\}$ such that $\sum_{i=1}^{N} w_i(x) = 1$, the final output of the MoE layer is a weighted combination of expert outputs based on these gating weights.

$$y = \sum_{i=1}^{N} w_i(x) \cdot E_i(x) \tag{2}$$

---

**Algorithm 1** Token-Complexity Based Routing for Mixture of Experts (MoE)

---

**Input:** Tokens $X$, Threshold $\alpha$, Lightweight Experts $E_{\text{weak}}$, Strong Experts $E_{\text{strong}}$, Router $R$
**Output:** Tokens $Y$

  1: Initialize token complexity $C \leftarrow \emptyset$
  2: Initialize output tokens $Y \leftarrow \emptyset$
  3: **for** each token $x \in X$ **do**
  4:    Compute complexity probability $P(C|x)$
  5:    **if** $P(C|x) \geq \alpha$ **then**
  6:      Route $x$ to strong experts $E_{\text{strong}}$
  7:      $y \leftarrow E_{\text{strong}}(x)$
  8:    **else**
  9:      Route $x$ to lightweight experts $E_{\text{weak}}$
10:      $y \leftarrow E_{\text{weak}}(x)$
11:    **end if**
12:    Append $y$ to $Y$
13: **end for**
14: Compute loss $\mathcal{L}_{\text{total}} = \mathcal{L}_{\text{task}} + \lambda_1 \mathcal{L}_{\text{aux}} + \lambda_2 \mathcal{L}_z$
15: Update Router $R$ using backpropagation with loss $\mathcal{L}_{\text{total}}$
16: **return** $Y$

---

MoE layer is placed to select the feed-forward network (FFN) within each transformer block, where the dense MLP is replaced with multiple parallel expert FFNs, usually following the self-attention sublayer. The model can be increasingly computationally demanding if the positioning is not correct, resulting in the model scaling up. A router module determines which experts to activate for each token, based on token representations. This allows the model to route simple tokens through lightweight experts and reserve high-capacity experts for complex tokens, achieving an effective balance between performance and efficiency. While designing a MoE architecture, another concept of load balancing is crucial, using a loss function. The loss function combines the primary task loss with auxiliary losses that balance the expert utilization. These auxiliary losses(typically in sparse MoE) are used to select a subset of experts per input. The overall loss can be explained using the following equation.

$$L_{\text{total}} = L_{\text{task}} + \lambda_1 L_{\text{importance}} + \lambda_2 L_{\text{load}} \tag{3}$$

Here, $L_{\text{task}}$ is the main objective such as cross-entropy for classification, $L_{\text{importance}}$ penalizes uneven routing weights across experts, and $L_{\text{load}}$ penalizes imbalanced token assignment across the expert pool. The hyperparameters $\lambda_1$ and $\lambda_2$ control the strength of these regularization terms. However, in our proposed framework, we achieved lower perplexity while using router-z loss proposed by Fedus et al. (2022) in Switch Transformer as it encourages uniform distribution of tokens with auxiliary loss with coefficient of 0.001 and 0.01 respectively Zoph et al. (2022). Although uniform routing uses all available experts regardless of any condition, we are using only the loss function to uniformly distribute the tokens based on their complexity within two lightweight and complex FFNs. The proposed framework uses the following equation for overall load balancing.

$$L_{\text{total}} = L_{\text{task}} + \lambda_1 L_{\text{aux}} + \lambda_2 L_z \tag{4}$$

However, traditional MoE systems often rely on simplistic routing heuristics such as top-k softmax. It lacks any awareness of token complexity, which might lead to suboptimal expert utilization and reduced generalization. Our proposed framework extends this concept by incorporating a threshold for determining a token complexity routing scheme that explicitly distinguishes between lightweight and complex tokens and activates appropriate FFNs accordingly. We have added the pseudocode for our expert selection based on token complexity in Algorithm. 1.

### 3.3 SPARSE MOE

Traditional MoE architectures used Dense MoE similar to ensemble learning and traditional neural networks, which used all experts simultaneously, and each expert contributed to the final decision. This increases computational overhead, and to make a solution to this problem, sparse MoE was

introduced, where a subset of FFN is activated. Traditional dense neural architectures activate all parameters for every input. Sparse MoE introduces conditional computation by activating only a small subset of experts for each input token. This design significantly reduces the computational overhead, allowing the model to scale to billions of parameters while keeping per-token computation relatively constant.

In our proposed framework, we have four experts, where two experts are designed to process higher complexity tokens, which might include math expressions and calculation, and the other two experts are designed for processing lightweight tokens, which are predictable and do not require much prior context compared to higher complexity tokens. We define the difference in the few-shot learning step while adding the neural network layer for predicting and passing the tokens to the appropriate FFN. We define the optimal threshold $\alpha$ from 0.478 to 0.5 to differentiate the difficulty level of the token while passing across the router. Tokens are distinguished as simple and complex within this range. The framework considers all the tokens as easy if the threshold is below 0.478 and all the tokens complex if the threshold is higher than 0.5. Hence, we have chosen this range as our optimal threshold range. The following equation is being used for making the Routing Decision controlling Token distribution based on the Token complexity C threshold, which is inspired by the cost threshold of RouteLLM Ong et al. (2024).

$$R_\alpha(\mathbf{q}) = \begin{cases} E_{\text{weak}}, & \text{if } P(C \mid \text{token}) < \alpha \\ E_{\text{strong}}, & \text{if } P(C \mid \text{token}) \geq \alpha \end{cases} \tag{5}$$

## 4 EXPERIMENTAL RESULTS

We analyze the perplexity for different numbers of active layers before deploying the router for an optimal number of active parameter configurations on the models' perplexity. Table 2 shows the perplexity of both Mistral and LLaMA-2 models for two configurations of active transformer layers for both training and inference. We evaluate the perplexity on the wikitext-v-3 dataset. The models show lower perplexity of 16.6 for Mistral and 7.40 for LLaMA-2 when we are fine-tuning 6-12 layers rather than 8-12 layers, which indicates the models achieve optimal stability when we are unfreezing more layers for training and inference. This suggests that activating a broader context of mid-to-upper layers improves parameter utilization and achieves an optimal prediction stability.

### 4.1 ABLATION STUDY

We evaluate the effectiveness of the proposed router-based model in Table. 3. Table. 3 shows the results for four benchmarks with different threshold values with and without adding the router framework to the base Mistral and Llama model for ablation study. We also show the number of tokens are being used for fine-tuning the router and how tokens are being distributed depending on the threshold value. All the values are in percentages in this table except token complexity, cost (no. of parameters fin-tuned), router. It shows the performance for the base Mistral-7B and Llama-2 model and router-based model's performance while evaluating on the same benchmarks. While evaluating on GSM8K (2-shot), our router achieves 58.4% accuracy with Mistral-7B and 82.6% accuracy with LLaMA-2-7B base. We achieve approximately 12% higher accuracy while using Llmama-2 as the base model and maintaining the threshold value as 0.5. From the analysis, we can observe that our routing mechanism significantly enhances logical reasoning in models. On the MMLU benchmark, our router surpasses base with Mistral-7B, achieving 74.8% while the threshold is 0.482, whereas the base Mistral model has 70.6% accuracy. With proposed router based LLaMA-2-7B, it achieves 56.4% with the base model being 45.3%. This demonstrates that routing tokens considering the complexity level can improve the generalization capability of the model on broad subject matter tasks and better utilize model capacity for knowledge-rich inputs. Similarly, in the open-domain QA benchmark Natural Questions, evaluated under a 5-shot prompting setup, the proposed framework improves over base model results across both models. For Mistral-7B, the accuracy improves from 28.8% to 31.5%, while for LLaMA-2-7B, it increases from 26.0% to 30.2%. Such performance gain validates that the token distribution is appropriate, which yields better performance, and token-level complexity awareness helps improve factual grounding by allocating expert capacity to tokens requiring deeper semantic modeling as well. We show token-distribution in Token Complexity (Complex vs. Simple tokens ratio)based on different threshold values for a smaller subset of benchmarks

to how optimal thresholding is improving performance. However, while evaluating on MBPP with a 3-shot prompting setup benchmark, with Mistral-7B, our router-based method achieves slightly less performance compared to the base model with 47% accuracy with threshold of 0.5, whereas the base model has 47.5% accuracy. For the Llama-2 base model, we see a performance drop of approximately 5% while using the proposed router. This can indicate that the effectiveness of expert routing for code generation may depend heavily on model-specific inductive biases or alignment quality for programming tasks.

## 4.2 COMPLEXITY ANALYSIS

In Table 4, we show the evaluation time complexity per benchmark while the threshold value was set to 0.5, which can help understand the latency and inference time of the proposed framework. It presents the evaluation time in GPU hours across four benchmark tasks for both Mistral and LLaMA-2, for both the base model and our proposed router-based approach, to show that while improving the performance, our framework also requires similar inference time while using an external routing system. For Mistral-7B, our method increases the average GPU hours slightly from 3.28 to 3.63, with the largest increase observed on MBPP from 3.5 to 4.34, on which the proposed framework's performance decreases as well. Similarly, for LLaMA-2, the average GPU hours rise from 3.81 to 4.07, with GSM8K and Natural Questions showing marginal increases. This indicates that while our method introduces additional routing computation, it maintains competitive efficiency across various evaluation settings.

Similarly, in Table 5, we report the iteration-level evaluation time for each inference, such as time per input instance for both Mistral and LLaMA-2 base models under base model configuration and our proposed difficulty-aware routing framework. For the Mistral model, our router introduces a moderate increase in iteration time across all tasks. On GSM8K, inference time rises from 1.38s to 2.04s, and on MMLU from 3.95s to 4.06s. The increase is more noticeable on MBPP from 5.08s to 7.29s and for Natural Questions from 5.6s to 6.84s, reflecting the additional overhead of routing and computing with specialized FFNs. Despite this increase, the improved accuracy (as shown in Table 1) justifies the trade-off in many use cases where performance is critical. For LLaMA-2, the impact on runtime is similar. Evaluation on GSM8K shows an increase from 8.2s to 10.28s, while MMLU increases from 1.16s to 1.30s. However, for MBPP and Natural Questions, the per-iteration runtime remains almost similar, indicating that the routing overhead is offset by efficient expert usage. The results demonstrate that while our approach incurs a modest increase in inference time per iteration, particularly on computationally heavy datasets like MBPP and GSM8K, the trade-off yields significant gains in accuracy and efficiency of computation allocation. The router's lightweight design ensures that the runtime remains within practical deployment thresholds, especially for real-world tasks where token-wise adaptability improves overall accuracy. We also show the average iteration time, which also shows a minimal increase in inference time, suggesting an optimal trade-off between complexity and accuracy.

## 4.3 PERFORMANCE ANALYSIS

Finally, in this section, we analyze the models' performance with the proposed router and compare with the state of the art technique DynaMoE on the other common benchmarks. For performance evaluation, we include Mixtral-$8 \times 7B$ model which is another MoE-based SOTA model where our router can be integrated. In Table 6, we show the accuracy, complexity distribution, and fine-tunable parameters for base model, SOTA routing technique and our proposed routing framework applied to the Mistral-7B, Mixtral-$8 \times 7B$ and LLaMA-2 models across seven benchmarks for different threshold values. The results from DynaMoE is collected from Nishu et al. (2025) with threshold value of 0.9. We are using similar threshold range for evaluating the performance of proposed method on ARC-e/c, LAMBADA, PIQA, WinoGrande, SciQ, HellaSwag. However, for these benchmarks, we are getting higher accuracy for the threshold value of 0.482. DynaMoE requires fine-tuning of 10B tokens which we are considering as their cost. We are fine-tuning 210M parameters of the base model for the router framework while using Mistral-2 and Mixtral-$8 \times 7B$ as base and 33.6M parameters while using Llama-2 as the base model similar to the previous ablation study. While evaluating the benchmarks of Table. 6, we observe the base Mistral model shows better performance compared to DynaMoE and proposed framework. However, it requires processing of higher number of tokens compared to the proposed framework which can be considered as a trade-off between cost

and accuracy while evaluating performance using comparatively less challenging and smaller scale benchmarks. Our proposed framework performs better than DynaMoE with comparatively lesser cost for all the benchmarks with threshold value of 0.482 while using the Mistral-2 as base.

We also see the base model outperforms the proposed framework for ARC-e, PIQA, and Wino-Grande (results collected from Nishu et al. (2025)). The proposed routing at threshold 0.5 with base as Mistral-2, achieves 71.0% on ARC-e, 82.0% on PIQA, and 91.0% on HellaSwag. If th thrshold is lower to 0.478, we see further reduction in complex-token usage which also leads to accuracy drop, particularly on ARC-e (68.0%) and PIQA (66.0%). For LLaMA-2, the base model performs with better accuracy on ARC-e (75.1%) and LAMBADA (71.5%) but at full token cost. The proposed routing method at threshold 0.5 achieves comparable ARC-e accuracy (72.0%) while drastically reducing token usage to 33.6M. Similar to Mistral, lowering the threshold to 0.478 yields the lowest token cost but also has a noticeable accuracy decline, especially for PIQA (66.0%) and ARC-c (38.0%). Hence, we can see, the results highlight a clear cost–accuracy trade-off and higher thresholds retain more complex tokens, preserving accuracy at the expense of cost, while lower thresholds significantly reduce token usage but may compromise performance across both models with the proposed routing system. Similarly, for evaluating the performance of Mixtral-$8 \times 7B$ with the proposed router, accuracy is comparatively higher than the router-based Mistral and Llama for most of the benchmarks. Although, we observe improvement is results while using the proposed router with Mixtral, there is no significant difference in accuracy for varying thresholds. Table. 6, also shows the token distribution for varying threshold values for a subset of questions from the benchmark which helps to understand the optimal distribution of tokens based on threshold value.

## 5 LIMITATIONS

While our proposed routing framework demonstrates higher accuracy compared to the base model and computational complexity for most of the benchmarks, it has several limitations. The framework relies on token-level difficulty labels for training the router, which are derived through heuristics of an MLP layer with few-shot examples. As we are using 2-shot for fine-tuning the router, these may not fully capture contextual difficulty, especially across diverse domains. Although our routing mechanism is lightweight, it introduces additional computation during inference. The framework is integrated and evaluated with two decoder-only base models rather than any encoder-only or encoder-decoder architecture, which also lack generalizability. While evaluating the framework's performance on programming problems, the routing system performs inadequately as it fails to distinguish between mathematical problem-related tokens and programming problem-related tokens, and requires additional logical fine-tuning for token differentiation to improve the accuracy.

## 6 CONCLUSION AND FUTURE WORKS

Our proposed approach contributes to the complexity aware routing scheme for flexible and efficient transformer inference. In this work, we proposed a routing framework that distributes tokens based on their difficulty level using an NN layer and few-shot learning. It enhances transformer-based language models' performance by selectively activating lightweight and strong FFN pathways for different tokens within each transformer block based on heuristic token complexity computation and route computation accordingly. It outperforms state-of-the-art base model performance for several benchmarks while evaluating on eleven diverse language benchmarks. The framework maintains efficient inference behavior and reduces perplexity with optimized configurations of 6-12 active parameters. Our approach can be considered for a resource-constrained environment as it improves the accuracy with minimal increase in computational overhead. In the future, we will add more base models of diverse architecture for router integration and evaluate performance. We are creating a routing threshold with binary expert selection in this research and in future, we would create experts which can distinguish between several levels of complexity and distribute the tokens depending on their levels for better accuracy and distribution of tokens within the experts.

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
