# OpenReview forum: "Token-Complexity based Routing Technique within Mixture of Experts Architecture for Large Language Model"
_ICLR.cc/2026/Conference — Submitted to ICLR 2026_

### Official Review · Reviewer_HMXo · 2025-10-17

**Soundness:** 3
**Presentation:** 1
**Contribution:** 2
**Rating:** 4
**Confidence:** 3

**Summary:**

this paper proposes a dynamic routing framework based on token complexity.
The framework trains a router using a few-shot classification objective and a surrogate neural network layer. It dynamically assigns tokens to either lightweight FFNs or strong FFNs based on the tokens’ semantic complexity. This complexity is calculated using features such as the hidden states of the prediction layer and the hidden states of the attention layer. For training optimization, the framework uses a total loss function that integrates task loss and load balancing loss.

**Strengths:**

The idea of routing different tokens based on their difficulty level is interesting.

**Weaknesses:**

1 The paper title on the OpenReview page is inconsistent with that in the main paper.

2 Typo: There is a duplicate "load" in the figure caption of Figure 1 ("optimal load load balancing").

3 In the experiments, the number of experts is only set to 4, and there is no exploration of more experts (e.g., 8). This means the generalizability of the proposed method needs to be verified.

4 The ratio of lightweight experts to strong experts is 1:1, and there is no analysis on how different ratios would affect the results.

5 In the experiments, most comparisons are between the proposed method and the base model, with a lack of comparisons against other router schemes.

6 The threshold α=0.478–0.5 only reflects performance degradation, and there is no analysis of the relationship between the threshold and token complexity distribution (e.g., how the proportion of complex tokens in different benchmarks affects the optimal threshold).

7 In Table 6, unlike the other two models, Mixtral-8×7B shows little fluctuation in accuracy under different thresholds. However, the authors do not explain the reason for this, which deserves in-depth analysis.

**Questions:**

see weakness

---

> ### Author Response · Authors · 2025-11-25
>
> Dear Reviewer, Thank you so much for your reviews. We have tried to address the weaknesses of the paper in the following manner -
>
> 1. We have corrected the title within the manuscript according to the OpenReview page.
> 2. We have corrected the typos and rewritten the manuscript based on the suggestions.
> 3. While using our proposed framework, we already have additional inference and memory costs. We are using 4 experts and considering 8 experts will introduce more cost, which will not be efficient even if accuracy gets higher and it would have been best if we could maintain this efficiency using 2 experts rather than even 4 in our framework. For the issue of generalizability, we evaluated the performance on various benchmarks and integrated the router with three open-source models to ensure it.
>
> 4. We can experiment with different ratios but it will change the architecture entirely, and the research will deviate from the main objective of the paper, to see how using a threshold to distribute tokens within experts' work. But the idea to change the ratio is very interesting, and we would experiment with it in the future.
>
> 5. We have added results of DynaMoE, which is the state-of-the-art technique and has the most efficient performance compared to any current MoE-nased implementations, achieving decent accuracy with reasonable cost. For most of the benchmarks, we achieved a better score than DynaMoE. Hence, we did not include any other MoE-based routers for comparison. In the future, we can include results from different routers to show the performance difference as well if needed. For the tables, we do not have such comparison, we do not have those exact metrics within the corresponding paper to compare with ours, and it might not be a fair comparison if we added those. Hence, we added results with and without the proposed router to see the difference.
>
> 6. As we mentioned in our previous responses regarding this comment, we wanted to try a wide range of thresholding. But below and above this threshold range, either one of the experts (either light expert or strong expert) was receiving zero tokens for processing. Hence, we had to choose this range for optimal token distribution while keeping the efficiency stable. Different benchmarks have different numbers of complex and easy tokens and depend on the threshold value as well. Besides, our results also show that a specific threshold does not outperform for all the benchmarks, but it shifts depending on the task and the complexity of the benchmark.
>
> 7. The router affects the results due to its distribution while using a comparatively larger model Mistral-8×7B. We have added more explanation while analyzing the results in the manuscript.

---

### Official Review · Reviewer_9kKX · 2025-10-28

**Soundness:** 1
**Presentation:** 1
**Contribution:** 1
**Rating:** 2
**Confidence:** 4

**Summary:**

In this submission, the authors introduce a routing mechanism of Mixture-of-Experts (MoE) models. The method aims to dynamically allocate tokens based on their estimated complexity: easy tokens are assigned to lightweight FFNs, and complex ones are routed to strong FFNs.  The authors state that this dynamic routing is implemented by a router trained with a “few-shot classification objective“ and a “surrogate neural network layer“. They experiment with Mistral-7B, and Llama2-7B demonstrates improvement on a series of benchmarks, including GSM8K, MMLU, and HellaSwag.

**Strengths:**

The motivation is quite straightforward and intuitive: tokens with different complexity should not be allocated with the same computation budget.

**Weaknesses:**

The paper suffers from severe flaws when describing the proposed method, the experiments, making the core method and the whole paper hard to understand and the results untrustworthy.

**Weakness 1 Severe Clarity Issue:** The paper fails to explain what its method and how it is applied.

- The paper applies its method to dense models (llama-2 and mistral) but does not explain the “MoEfication“ process. For example, what’s the new lightweight and strong FFN architecture and how their parameters initialized. This fundamental detail could help reader better understand the paper. The paper also mentions a "surrogate neural network layer" (Fig 1) and a "Router R" (Algorithm 1) but does not explain them well.

- The Eq. 1 and algorithm 1 is also confusing. For example, The paper names $D_{ctx}(t)$ as "contextual difficulty features," but the paper does not define what these features are or how they are computed. The paper repeatedly mentions the "few-shot classification objective" (Abstract, Sec 5). But it is unclear what this task is, where the "few-shot" examples come from (are they from the heuristic?), or how this objective is used to train the router. Furthermore, the paper does not *state its own fine-tuning settings* (dataset, total token count, optimizer, etc.).


**Weakness 2 Questionable experiment results:** the paper presents problematic experiment results.

- **Skeptical Baseline Results:** Table 3 reports a baseline accuracy for Llama-2-7B on GSM8K (5-shot) of **82.6%**. This number is much higher and inconsistent with established literature.

- **Misleading "Cost" Metric:** In Table 6, the "Cost" column is misleading. As per Section 4.3, the cost for the proposed method (`~210M`) is the **number of fine-tuned parameters** (if I understand right), while the cost for DynaMoE (`10B`) is the **number of fine-tuning tokens**. This is highly confusing.

- **No performance gains on several tasks:** Table 6 shows the *base* Mistral-7B model **outperforming** the proposed method on several benchmarks (e.g., 80.2% vs 75.0% on ARC-e; 96.4% vs 86.0% on SciQ).


**Weakness 3 poor presentation**

In addition to present the method and experiments poorly, there are other minor things. For example, the title does not match the abstract ("Efficiency" vs. "Complexity"). And what is this Mistral-2 model (line 434)?

**Questions:**

Here are my questions for authors about the proposed method:

1. What is the definition of token complexity discussed in this paper? The paper seems give two definitions: a heuristic rule-based definition (line 258) and a learned function (Eq.1). How are they related?

2. What are the "contextual difficulty features" (`D_ctx(t)`) and "token type embedding" (`T_type(t)`), and how are they computed?

3. What is the "few-shot classification objective"? Please detail the task and the data, the number of "shots," and how this objective is used to train the router.

4. What is the *exact* architectural difference between your parallel "lightweight" and "strong" experts and the "nested" experts used by DynaMoE?

5. Please state your full fine-tuning details: what dataset, how many total tokens, what optimizer, and fine-tuning hyper-parameters were used? This is necessary to make a fair comparison to DynaMoE's 10B token cost.

**Details Of Ethics Concerns:**

N / A

---

> ### Author Response · Authors · 2025-11-27
>
> Dear Reviewer, Thank you so much for your review. We have tried to address the questions along with the concerns in the following manner –
>
> 1. We understand the confusion here. We used few-shot examples before using Equation 1 to learn the MLP layer. We mentioned the equation before the few-shot specification which we interchanged to enhance the readability. We specified math expressions, code syntax etc. within the few-shot (2-shot) example. Then the model starts learning from the specified function within the MLP sub-layer.
> 2. Contextual tokens are basically for getting the local context surrounding a specific token t that influences its processing difficulty. It is calculated with the attention weights received by token t from its surrounding tokens using the local context window and cosine similarity and few-shot learning to distinguish between a complex and simple token depending on a assigned threshold value. Finally, we get the the token type embeddings from the learning represented as T_type(t) based on its corresponding complexity. We tried to elaborate on the technique in the manuscript as well to enhance readability.
> 3. The router is trained as a binary classifier that predicts whether a token should be routed to lightweight experts or strong experts. The classification objective is a cross-entropy loss over these two classes, combined with the auxiliary and router-z losses described in Equation 4 similar to the typical MoE-based models load balancing. We used 2-shot learning for the router as we are training it for distinguishing between complex and easy tokens.
> 4. Our architecture for the different experts varied in size mostly as mentioned in previous responses. We use 4x expansion when we use the strong experts for handling the complex tokens and 2x expansion for lightweight experts. DynaMoE creates experts by partitioning the original dense MLP weight matrices into nested subnetworks of varying sizes. While differentiating from the architecture of DynaMoE, we use a threshold while routing the tokens within the MLP sub-layer along with reduced hidden dimension for lightweight experts and full hidden dimensions for strong experts. We elaborated more on the architecture in the methodology section.
> 5. We understand the issue with the cost comparison of the proposed method with DynaMoE which was very helpful for us to work further on the project. We mentioned the parameter size fine-tuning as our cost in the manuscript which creates a discrepancy with the number of token fine-tuned cost. Hence, while calculating the number of tokens, we are fine-tuning, in the proposed framework is higher than 10B tokens. We will mention this issue as our limitation and in future we will work on reducing the cost and making a fair comparison with DynaMoE.
>
> Besides, these, we corrected the writings such as adding correct baseline for GSM8k and what might be possible reasons for lower performance of the proposed method compared to the base model in the manuscript.

---

### Official Review · Reviewer_rKMv · 2025-10-30

**Soundness:** 2
**Presentation:** 2
**Contribution:** 2
**Rating:** 4
**Confidence:** 4

**Summary:**

The paper presents a complexity-aware routing framework designed to enable flexible and efficient Transformer inference. It introduces a neural routing module that estimates token difficulty and dynamically allocates computation by activating either lightweight or strong expert feed-forward pathways. The routing function is trained via a small neural network with few-shot learning, allowing adaptive expert selection without full model retraining. The approach is lightweight, model-agnostic, and easily integrable into existing LLM architectures. Experimental results on multiple base models, including Mistral and LLaMA, demonstrate favorable accuracy–compute trade-offs across various configurations.

**Strengths:**

- Well-motivated problem formulation: Clearly identifies inefficiencies in uniform token processing and motivates complexity-aware routing as a principled way to allocate compute based on token difficulty.
- Lightweight and modular design: The proposed routing layer introduces minimal overhead, requires few-shot calibration, and integrates seamlessly with existing LLM architectures without retraining.
- Dynamic compute allocation: Effectively balances efficiency and accuracy by selectively activating lightweight or strong FFN pathways per token, demonstrating adaptive computation within standard Transformer blocks in dense or MoE base model.
- Empirical validation across multiple base models: Evaluations on Mistral and LLaMA confirm consistent accuracy–compute trade-offs, highlighting the method’s generality and practical applicability.

**Weaknesses:**

Several introduced concepts lack precise definitions or sufficient experimental justification for their use.

- Insufficient explanation of difficulty labeling: The process for deriving token difficulty labels is not clearly described. Clarify whether these labels are generated automatically, depend on model confidence, or require annotations from the fine-tuning dataset. The quality of difficulty labels would impact the actual routing decision being learnt. So, without much validation of the ground truth labels, the proposed method would not be broadly applicable.

- Unclear expert architecture details: The paper does not clearly specify the architectural differences between the strong and weak experts and how that will be derived based on the underlying structure of the MLP layers of any new base LLM.

- Ambiguous token difficulty thresholds:
The chosen thresholds (≤0.48 for easy tokens, >0.5 for complex ones) appear arbitrary, with a very narrow margin separating the two classes. The authors should justify this decision.

- Undefined computation of $L_{importance}$ loss: Provide its explicit mathematical definition and intuition, and clarify how it interacts with the main training objective.

- Minor presentation and formatting issues:
The paper contains inconsistent quotation formatting (e.g., “easy” and “hard” tokens) and unclear table descriptions. Improve clarity by:
expanding table captions to be self-contained, defining metrics like Token Complexity (Table 3) and iteration-level evaluation (Table 5).
Ensuring Table 4’s description is coherent and aligned with the experimental setup.

**Questions:**

The writing of the paper needs a revision in order to make it more polished and coherent.

---

> ### Author Response · Authors · 2025-11-25
>
> Dear Reviewer, Thank you so much for your review. We have tried to address the weaknesses mentioned in the paper in the following manner –
>
> 1. We understand the issue with the learning of the complexity of the router. Router is a part of the transformer layer and each layer we work is trained with the few-shot examples we add. Few shot examples include math expressions, code syntax, and words that are complex while tokenizing. We added a separate paragraph in the Methodology section regarding this complexity analysis of tokens.
>
> 2. Architecture for the experts vary in size mostly. We use 4x expansion when we use the strong experts for handling the complex tokens and 2x expansion for lightweight experts. We will try to add an illustration that shows the architecture instead of the general architecture we have in Figure. 2 for lightweight and strong expert FFNs along with the fine-tuning specifications in the draft.
>
> 3. We understand that confusion for our thresholding range. As we mentioned in our previous responses as well regarding this comment, we wanted to try a wide range of thresholding. But below and above this threshold range either one of the experts (either light expert or strong expert) was receiving zero tokens for processing. The number of tokens being processed we reported in the paper are basically originated when we were analyzing errors in our results with a random threshold. Only optimal range where both lightweight and strong experts were receiving a number of tokens to process was 0.478-0.5 depending on our architecture. Hence, this range had to be chosen for complexity analysis.
>
> 4. We used the loss L_importance to describe how MoE-based router use different loss function for load balancing and we used the Equation-3 to describe how load balancing works. This was not used in our implementation. As we mentioned in line 309, we are using the z-router in the proposed framework which is in Equation-4 and proposed by Fedus et. al and simply integrated within the router.
> Reference :
> 1.	William Fedus, Barret Zoph, and Noam Shazeer. Switch transformers: Scaling to trillion parameter models with simple and efficient sparsity. Journal of Machine Learning Research, 23(120):1–39, 2022.
>
> 5. We corrected the formatting issues and rewritten based on your suggestions in the draft.

---

### Official Review · Reviewer_3Bi6 · 2025-11-01

**Soundness:** 2
**Presentation:** 1
**Contribution:** 2
**Rating:** 4
**Confidence:** 2

**Summary:**

This paper proposes a token‑complexity‑based dynamic routing mechanism for improving Mixture‑of‑Experts (MoE) architectures in large language models. Unlike conventional fixed top‑k routing, the method employs a lightweight neural router trained with few‑shot supervision to estimate each token’s complexity and dynamically assign it to either “lightweight” or “strong” feedforward expert networks (FFNs) based on a learned threshold α. Integrated with models such as Mistral‑7B, Mixtral‑8×7B, and LLaMA‑2‑7B, the framework achieves up to 12% accuracy improvements across benchmarks including GSM8K, MMLU, and HellaSwag, while maintaining stable computational cost. Overall, this work provides an efficient, scalable approach to enhance expert utilization and inference efficiency in transformer‑based large language models.

**Strengths:**

- The paper explores an interesting direction by integrating token‑level complexity awareness into MoE routing, which conceptually extends prior adaptive expert frameworks like DynaMoE and Flextron toward a more fine‑grained per‑token perspective.

- The idea of using a few‑shot supervised router with a tunable complexity threshold is a creative attempt to make routing both data‑efficient and flexible, showing potential for broader applicability in resource‑constrained LLM deployment.

- The authors conduct comprehensive empirical evaluations across diverse benchmarks, demonstrating consistent, though moderate, improvements that illustrate the promise of their complexity‑based routing strategy for enhancing expert utilization efficiency.

**Weaknesses:**

- The overall writing quality of the paper requires substantial improvement. First, the use of citations is incorrect and inconsistent — the authors do not properly distinguish among \cite, \citep, and \citet, as seen for example at lines 131 and 140. Second, the formatting of the tables does not follow the ICLR template guidelines, since the table borders should open on both sides. In addition, some figures, such as Figure 2, are not visually appealing — there is excessive spacing between subfigures, and the font size is too small. Moreover, table references (e.g., “Table. 3.” at line 357) are not hyperlinked, making them unclickable. Overall, the writing and presentation quality of the manuscript still need significant improvement.

- The proposed token-complexity-based routing concept overlaps substantially with prior adaptive MoE systems such as DynaMoE [1] and Flextron [2], which also route tokens or features based on difficulty or task semantics. The few-shot supervised router and complexity threshold provide some variation, but the paper does not sufficiently explain what theoretical gap this fills beyond existing “dynamic routing” approaches (e.g., Token-level Load Balancing). The paper should clarify how its method differs in the design of gating functions or optimization objectives, ideally supported by quantitative or theoretical evidence—such as improved load balance entropy, lower variance in expert utilization, or convergence guarantees—to substantiate novelty claims.

- The definition of token complexity relies on heuristics—for instance, labeling mathematical or programming symbols as “complex” and common words as “simple.” This rule-based labeling can cause poor generalization across domains because it ignores probabilistic uncertainty or contextual diversity. A more rigorous quantitative measure such as token entropy, surprisal, contextual dependency length, or prediction variance should replace heuristic rules to better reflect real difficulty. Moreover, verifying the correlation between estimated complexity C(t) and empirical uncertainty measures (e.g., perplexity) would strengthen the validity of the metric. If such signals were integrated into the router, the routing decisions could become more consistent across tasks.

- The experiments evaluate a wide set of benchmarks but lack the necessary depth to isolate where the performance improvements originate. The study omits comparisons to recent dynamic router baselines such as RouteLLM and RouterBench.


**Reference:**

[1] From dense to dynamic: Token-difficulty driven
moefication of pre-trained llms.

[2]  Flextron: Many-in-one flexible large language model.

**Questions:**

- Could the authors clarify how their token‑complexity‑based router fundamentally differs from prior adaptive MoE systems such as DynaMoE or Flextron?
- How was the heuristic definition of token complexity validated, and does it correlate with quantitative measures like entropy or surprisal?
- What is the sensitivity of the model’s performance to the chosen routing threshold \( \alpha \), and how was this value determined?
- Why were recent baselines like RouteLLM and RouterBench excluded from the experimental comparisons?
- Can the authors provide a breakdown of the additional inference cost introduced by the routing mechanism?

---

> ### Author Response · Authors · 2025-11-25
>
> Dear Reviewer, Thank you so much for your review. We tried to rewrite the paper based on your reviews for correcting the Table formatting and Figures to enhance the readability. We addressed the following concerns as follows –
>
> 1. As we know DynaMoE basically uses similarity score to find a specific token’s complexity. It derives token difficulty from similarity-based heuristics without having any access to supervised or semantically grounded labels. As a result token difficulty is indirectly approximated and fixed at deployment time. On the contrary, our proposed router shifts routing decisions based on a threshold value demonstrating a complexity-aware behavior which is not present in prior systems.
>
> Now coming to the Flextron, it uses a surrogate model to approximate downstream loss, but it does not perform token-level complexity-aware routing inside the MLP pathway. Our proposed router, computes per-token complexity scores dynamically thereby choosing lightweight or strong experts depending on the score which allows the routing decision to be adaptive across different tasks and input. Our results also show that, a specific threshold does not outperform for all the benchmarks, but it shifts depending on the task of the benchmark which adds another novelty to the framework.
>
> 2. The heuristic definition of token-complexity was shown using the token surprisal as shown in Table. 2 with perplexity score. Hence, we can say the heuristic token-complexity labels were validated through both quantitative and task-level analyses. First, we measured the alignment between our complexity predictions and statistical difficulty measures such as token surprisal and decide how many layers we should fine-tune with the router. Table 2 shows that integrating the complexity-aware router reduces perplexity for both Mistral and LLaMA-2, indicating that the complexity classifier captures stable difficulty patterns when 6-12 transformer layers were fine-tuned. Also, we have Complex-Simple token-distribution patterns in Tables 3 and 6 demonstrating allocation across thresholds and models, supporting the validity of our complexity definition. However, we will work for creating a specific definition for the complexity where we can show a correlation between the perplexity and the results we got after evaluation to validate the complexity definition.
>
> 3. The thresholding of the routers was interesting. We wanted to try a wide range of thresholding. However, below and above this threshold range either one of the experts (either light expert or strong expert) was receiving zero tokens for processing. The number of tokens being processed we reported in the paper are basically originated when we were analyzing errors in our results with a random threshold. Only optimal range where both lightweight and strong experts were receiving a number of tokens to process was 0.478-0.5 depending on our architecture.
>
> 4. As we mentioned in the Related Works, RouteLLM is not a framework of Mixture of Expert. They simple use a threshold on the surface level to find an optimal cost efficient LLM to route the question. Hence, there is no option or point to compare their results with ours. Same goes for RouterBench. We are inspired to develop a MoE-based architecture from these frameworks but comparing their surface-level MoE architecture with a transformer-level architecture might not make sense.
>
> 5. We were expecting a higher inference cost while integrating the router. However, we have somewhat reasonable cost for both inference and memory while using it. For breaking down the cost, we can refer to the Table. 4 and 5, where the base model and proposed model shows the cost difference which indicates the additional cost which comes with the router integration.

---

> > ### Comment · Reviewer_3Bi6 · 2025-11-27
> > **Thanks for Authors’ Rebuttal**
> >
> > I thank the authors for their detailed response and for their efforts to improve the manuscript's formatting and presentation. I have read the rebuttal carefully. While some points are clarified, I still have concerns regarding the stability of the proposed method and the validation of the core concepts. I have decided to maintain my current score for now. Please find my follow-up comments and questions below:
> >
> > **1. Regarding Threshold Sensitivity and Stability (Major Concern)**
> >
> > In the rebuttal, the authors state: *"Only optimal range where both lightweight and strong experts were receiving a number of tokens to process was 0.478-0.5 depending on our architecture."* This admission raises a significant concern about the robustness of the proposed method. A functional hyperparameter range of only `0.022` (0.478 to 0.500) suggests that the router's output probabilities are extremely concentrated or that the system is highly brittle. In a robust MoE system, we would expect the router to learn a distribution that allows for a wider, more stable range of thresholds, or to use a top-k mechanism that avoids such collapse.
> > *   **My Question is:** Why is the effective threshold range so narrow? Does this imply that the router is not effectively learning to separate "easy" and "hard" tokens with high confidence (i.e., most sigmoid outputs are hovering near 0.5)? Did you apply any auxiliary load-balancing loss (like the one mentioned in Eq. 4) specifically to encourage a wider variance in the router's output probabilities?
> >
> > **2. Regarding the Definition and Validation of Token Complexity**
> >
> > The authors argue that Table 2 (perplexity reduction) validates the heuristic complexity labels. However, a reduction in overall model perplexity only proves that the *architecture* works, not that the *heuristic labels* (e.g., treating specific math symbols as "complex") correlate with actual statistical difficulty (surprisal/entropy).
> >
> > *   **My Question is:** I appreciate the honesty that you "will work for creating a specific definition." However, for the current submission, is there any quantitative evidence *directly* correlating your heuristic labels with the model's internal uncertainty? For example, before training the router, do the tokens you labeled as "complex" actually have higher loss/entropy in the base model compared to "simple" tokens? Without this, the definition remains arbitrary.
> >
> > **3. Regarding Baselines (RouteLLM/RouterBench)**
> >
> > I accept the authors' clarification that RouteLLM and RouterBench operate at the model/query level rather than the token level. Excluding them is acceptable given the architectural differences. However, this places more burden on the comparison with DynaMoE and the internal ablation studies to prove the method's efficacy.
> >
> > **4. Regarding Novelty vs. DynaMoE**
> >
> > The authors state that DynaMoE uses similarity heuristics while the proposed method uses a threshold-based complexity score. I am still not fully convinced that this represents a fundamental shift. Both approaches rely on heuristics to proxy difficulty. The primary difference appears to be the training signal (few-shot supervision vs. similarity). **Could you elaborate on why the few-shot supervision provides a superior signal for *routing* specifically? Does it lead to better expert specialization compared to similarity-based grouping?**
> >
> > I look forward to your response.

---

> > > ### Author Response · Authors · 2025-11-28
> > >
> > > Dear Reviewer, Thank you so much for your careful insights. We will try to address the issues clearly within the manuscript in the future. We tried to answer the questions as follows for each of the concerns –
> > >
> > > 1. Regarding Threshold Sensitivity and Stability
> > >
> > > We understand the issue with the narrow router range. It might not indicate router failure or probability miscalculation, in our opinion. It somewhat reflects the token distribution technique of the framework’s architecture and its decision mapping, which is tied to the few-shot training paradigm as well, where the router learns a conservative decision boundary rather than an overconfident hallucinated one. Even if we are using load balancing with a state-of-the-art z-router, we are getting token distribution as reported in the Table. 3 and 4, which is mostly because of the threshold, which also shows the effectiveness of the router. However, we understand the load balancing issue and will work on it further if this distribution is considered crucial.
> > >
> > > 2. Regarding the Definition and Validation of Token Complexity
> > >
> > > In the current submission, we do not have any direct correlation analysis to show the relationship between heuristic labels and the model’s internal uncertainty. For the experiment, we need to calculate the direct loss or entropy for tokens and see if complex tokens have higher loss and entropy than simple tokens, if we understand the suggestion correctly. We will try to add the correlation where we can show, before the integration of the router (in the base model), we have higher loss/entropy for complex tokens or not, and after router integration, how the correlation shifts or behaves to make a definition using this heuristic correlation analysis of tokens.
> > >
> > > 4.  Regarding Novelty vs. DynaMoE
> > >
> > > If we consider the fundamental difference of adding few-shot training (disregarding the thresholding), which originally increases the performance rather than the thresholding, then to answer the question, we can say that for most benchmarks, providing knowledge helps the framework to route the tokens within appropriate experts depending on the task. When we see lower performance while using the framework for some benchmarks, this indicates a generalization limitation that few-shot training might have. Hence, to answer the question, few-shot training can provide a superior signal for routing but will lack somewhat generalization capability, and thresholding is crucial with it for analyzing the token’s inherent complexity. Now, comparing to similarity-based techniques also might have the issue of a generalization issue, which thresholding the router and few-shot training combined can improve. We will further work on technical analysis on this concept and add quantitative analysis for a better understanding of the proposed router’s effect.

---

### Meta-Review · Area_Chair_7cVW · 2026-01-06

**Summary:**

Reviewers found the high-level idea of routing tokens based on difficulty intuitive, but raised serious concerns about clarity, robustness, and experimental rigor. Key aspects of the method—including the definition and validation of token complexity, the router objective, and the expert architecture—are insufficiently specified, making the approach hard to reproduce and trust. Reviewers also noted brittle behavior (e.g., an extremely narrow effective threshold range), questionable cost accounting, and inconsistent or weak comparisons to relevant dynamic routing baselines. While the rebuttal clarified some points, these core concerns remain unresolved, and the discussion does not support a defensible acceptance.

**Reviewer Concerns:**

The rebuttal addressed some presentation issues and clarified parts of the intended design, including formatting, high-level motivation, and partial explanations of the routing mechanism. However, core concerns remain outstanding, particularly regarding the definition and validation of token complexity, the narrow and brittle threshold behavior, unclear cost accounting, and insufficiently fair or consistent baseline comparisons. As a result, questions about methodological soundness and experimental credibility were not fully resolved.

**Reviewer Scores:**

Overall, the score distribution would remain largely unchanged and continue to lean below the acceptance threshold.

---

### Decision · Program_Chairs · 2026-01-26

Reject